# Bi-Objective Online Matching and Submodular Allocations

**Hossein Esfandiari**
University of Maryland
College Park, MD 20740
hossein@cs.umd.edu

**Nitish Korula**
Google Research
New York, NY 10011
nitish@google.com

**Vahab Mirrokni**
Google Research
New York, NY 10011
mirrokni@google.com

## Abstract

*Online* allocation problems have been widely studied due to their numerous practical applications (particularly to Internet advertising), as well as considerable theoretical interest. The main challenge in such problems is making assignment decisions in the face of uncertainty about future input; effective algorithms need to predict which constraints are most likely to bind, and learn the balance between short-term gain and the value of long-term resource availability.

In many important applications, the algorithm designer is faced with multiple objectives to optimize. In particular, in online advertising it is fairly common to optimize multiple metrics, such as clicks, conversions, and impressions, as well as other metrics which may be largely uncorrelated such as 'share of voice', and 'buyer surplus'. While there has been considerable work on multi-objective offline optimization (when the entire input is known in advance), very little is known about the online case, particularly in the case of adversarial input. In this paper, we give the first results for bi-objective online *submodular* optimization, providing almost matching upper and lower bounds for allocating items to agents with two submodular value functions. We also study practically relevant special cases of this problem related to Internet advertising, and obtain improved results. All our algorithms are nearly best possible, as well as being efficient and easy to implement in practice.

## 1 Introduction

As a central optimization problem with a wide variety of applications, online resource allocation problems have attracted a large body of research in networking, distributed computing, and electronic commerce. Here, items arrive one at a time (i.e. *online*), and when each item arrives, the algorithm must irrevocably assign it to an agent; each agent has a limited resource budget / capacity for items assigned to him. A big challenge in developing good algorithms for these problems is to *predict* future binding constraints or *learn* future capacity availability, and allocate items one by one to agents who are unlikely to hit their capacity in the future. Various stochastic and adversarial models have been proposed to study such online allocation problems, and many techniques have been developed for these problems. For stochastic input, a natural approach is to build a predicted instance (for instance, via sampling, or using historical data), and some of these techniques solve a dual linear program to learn dual variables that are used by the online algorithm moving forward [6, 10, 2, 23, 16, 18]. However, stochastic approaches may provide poor results on some input (for example, when there are unexpected spikes in supply / demand), and hence such problems have been extensively studied in adversarial models as well. Here, the algorithm typically maintains a careful balance between greedily *exploiting* the current item by assigning it to agents with high value for it, and assigning the item to a lower-value agent for whom the value is further from the distribution of 'typical' items they

have received. Again, primal-dual techniques have been applied to learn the dual variables used by the algorithm in an online manner [17, 3, 9].

A central practical application of such online algorithms is the online allocation of impressions or page-views to ads on the Internet [9, 2, 23, 5, 7]. Such problems are present both in the context of sponsored search advertising where advertisers have global budget constraints [17, 6, 3], or in display advertising where each ad campaign has a desired goal or a delivery constraint [9, 10, 2, 23, 5, 7]. Many of these online optimization techniques apply to general optimization problems including the online submodular welfare maximization problem (SWM) [20, 13].

For many real-world optimization problems, the goal is to optimize multiple objective functions [14, 1]. For instance, in Internet advertising, such objectives might include revenue, clicks, or conversions. A variety of techniques have been developed for multi-objective optimization problems; however, in most cases, these techniques are only applicable for offline multi-objective optimization problems [21, 26], and they do not apply to online settings, especially for online competitive algorithms that work against an adversarial input [17, 9] or in the presence of traffic spikes [18, 8] or hard-to-predict traffic patterns [5, 4, 22].

**Our contributions.** Motivated by the above applications and the increasing need to satisfy multiple objectives, we study a wide class of multi-objective online optimization problems, and present both hardness results and (almost tight) bi-objective approximation algorithms for them. In particular, we study resource allocation problems in which a sequence of items (also referred to as impressions) $i$ from an unknown set $I$ arrive one by one, and we have to allocate each item to one agent (for example, one advertiser) $a$ in a given set of agents $A$. Each agent $a$ has two monotone submodular set functions $f_a, g_a \colon 2^I \to R$ associated with it. Let $\mathcal{S}_a$ be the set of items assigned to bin $a$ as a result of online allocation decisions. The goal of the online allocation algorithm is to maximize two social welfare functions based on $f_a$'s and $g_a$, i.e., $\sum_{a \in A} f_a(\mathcal{S}_a)$ and $\sum_{a \in A} g_a(\mathcal{S}_a)$. We first present almost tight online approximation algorithms for the general online bi-objective submodular welfare maximization problem (see Theorems 2.3 and 2.5, and Fig. 1). We show that a simple random selection rule along with the greedy algorithm (when each item arrives, randomly pick one objective to greedily optimize) results in almost optimal algorithms. Our allocation rule is thus both very fast to run and trivially easy to implement. The main technical result of this part is the hardness result showing that the achieved approximation factor is almost tight unless P=NP. Furthermore, we consider special cases of this problem motivated by online ad allocation. In particular, for the special cases of online budgeted allocation and online weighted matching, motivated by sponsored search and display advertising (respectively), we present improved primal-dual-based algorithms along with improved hardness results for these problems (see, for example, the tight Theorem 3.1).

**Related Work.** It is known that the greedy algorithm leads to a $1/2$-approximation for the submodular social welfare maximization problem (SWM) [11], and this problem admits a $1 - 1/e$-approximation in the offline setting [24], which is tight [19]. However, for the online setting, the problem does not admit a better than $1/2$-approximation algorithm unless P= NP [12]. Bi-objective online allocation problems have been studied in two previous papers [14, 1]. The first paper presents [14] an online bi-objective algorithm for the problem of maximizing a general weight function and the cardinality function, and the second paper [1] presents results for the combined budgeted allocation and cardinality constraints. Our results in this paper improve and generalize those results for more general settings. Submodular partitioning problems have also been studied based on mixed robust/average-case objectives [25].

Our work is related to online ad allocation problems, including the *Display Ads Allocation (DA)* problem [9, 10, 2, 23], and the *Budgeted Allocation (AdWords)* problem [17, 6]. In both of these problems, the publisher must assign online impressions to an inventory of ads, optimizing efficiency or revenue of the allocation while respecting pre-specified contracts. The Display Ad (DA) problem is the online matching problem described above with a single weight objective [9, 7]. In the Budgeted Allocation problem, the publisher allocates impressions resulting from search queries. Advertiser $a$ has a budget $B(a)$ on the total spend, instead of a bound $n(a)$ on the number of impressions. Assigning impression $i$ to advertiser $a$ consumes $w_{ia}$ units of $a$'s budget instead of 1 of the $n(a)$ slots, as in the DA problem. For both of these problems, $1 - \frac{1}{e}$-approximation algorithms have been designed under the assumption of large capacities [17, 3, 9]. None of the above papers for adversarial models studies multiple objectives at the same time.

## 2 Bi-Objective Online Submodular Welfare Maximization

### 2.1 Model and Overview

For any allocation $\mathcal{S}$, let $\mathcal{S}_a$ denote the set of items assigned to agent $a \in A$ by this allocation. In the classic Submodular Welfare Maximization problem (SWM) for which there is a single monotone submodular objective, each agent $a \in A$ is associated with a submodular function $f_a$ defined on the set of items $I$. The welfare of allocation $\mathcal{S}$ is defined as $\sum_a f_a(\mathcal{S}_a)$, and the goal of SWM is to maximize this welfare. In the classic SWM, the natural greedy algorithm is to assign each item (when it arrives) to the agent whose gain increases the most. This greedy algorithm (note that it is an online algorithm) is $(1/2 + 1/n)$-competitive, and this is the best possible [15].

In this section, we consider the extension of online SWM to *two* monotone submodular functions. Formally, each agent $a \in A$ is associated with two submodular functions - $f_a$ and $g_a$ - defined on $I$. The goal is to find an allocation $S$ that does well on both objectives $\sum_a f_a(S_a)$ and $\sum_a g_a(S_a)$. We measure the performance of the algorithm by comparison to the offline optimum for each objective: Let $\mathcal{S}^{*f} = \arg\max_{\text{allocations } \mathcal{S}} \sum_a f_a(\mathcal{S}_a)$ and $\mathcal{S}^{*g} = \arg\max_{\text{allocations } \mathcal{S}} \sum_a g_a(\mathcal{S}_a)$. An algorithm $A$ is $(\alpha, \beta)$-competitive if, for every input, it produces an allocation $\mathcal{S}$ such that $\sum_a f_a(\mathcal{S}_a) \geq \alpha \sum_a f_a(\mathcal{S}_a^{*f})$ and $\sum_a g_a(\mathcal{S}_a) \geq \beta \sum_a g_a(\mathcal{S}_a^{*g})$.

A $(1, 1)$-competitive algorithm would be one that finds an allocation which is simultaneously optimal in both objectives, but since the objectives are distinct, no single allocation may maximize both, even ignoring computational difficulties or lack of knowledge of the future. One could attempt to maximize a linear combination of the two submodular objectives, but since the linear combination is itself submodular, this is no harder than the classic online SWM. Instead, we provide algorithms with the stronger guarantee that they are *simultaneously* competitive with the optimal solution for each objective separately. Further, our algorithms are parametrized, so the user can balance the importance of the two objectives.

Similar to previous approaches for bi-objective online allocation problems [14], we run two simultaneous greedy algorithms, each based on one of the objective functions. Upon arrival of each online item, with probability $p$ we pass the item to the greedy algorithm based on the objective function $f$, and with probability $1 - p$ we pass the item to the greedy algorithm based on $g$.

First, as a warmup, we provide a charging argument to show that the greedy algorithm for (single-objective) SWM is $1/2$-competitive. This charging argument is similar to the usual primal-dual analysis for allocation problems. However, since the objective functions are not linear, it may not be possible to interpret the proof using a primal-dual technique. Later, we modify our charging argument and show that if we run the greedy algorithm for SWM but only consider items for allocation with probability $p$, the competitive ratio is $\frac{p}{1+p}$. (Note that a naive analysis would yield a competitive ratio of $p/2$, since we lose a factor of $p$ in the sampling and a factor of $1/2$ due to the greedy algorithm.)

Since our algorithm for bi-objective online SWM passes items to the 'first' greedy algorithm with probability $p$ and passes items to the second greedy algorithm with probability $1 - p$, the modified charging argument immediately implies that our algorithm is $(\frac{p}{1+p}, \frac{1-p}{2-p})$ competitive, as we state in Theorem 2.3 below. Also, using a factor-revealing framework, assuming $NP \neq RP$, we provide an almost tight hardness result, which holds even if the objective functions have the simpler 'coverage' structure. Both our competitive ratio and the associated hardness result are presented in Figure 1.

### 2.2 Algorithm for Bi-Objective online SWM

We define some notation and ideas that we use to bound the competitive ratio of our algorithm. Let *Gr* be the greedy algorithm and let *Opt* be a fixed optimum allocation. For an agent $j$, and an algorithm *Alg*, let $Alg_j$ be the set of online items allocated to the agent $j$ by *Alg*; $Opt_j$ denotes the set of online items allocated to $j$ in *Opt*. Trivially, for any two agents $j$ and $k$, we have $Alg_j \cap Alg_k = \emptyset$.

For each online item $i$ we define a variable $\alpha_i$, and for each agent $j$ we define a variable $\beta_j$. In order to bound the competitive ratio of the algorithm *Alg* by $c$, it suffices to set the values of $\alpha_i$s and $\beta_j$s such that 1) the value of *Alg* is at least $c\left(\sum_{i=1}^n \alpha_i + \sum_{j=1}^m \beta_j\right)$ and 2) the value of *Opt* is at most $\sum_{i=1}^n \alpha_i + \sum_{j=1}^m \beta_j$.

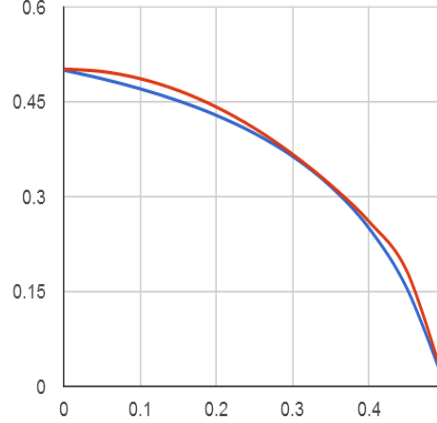

Figure 1: The lower (blue) curve is the competitive ratio of our algorithm, and the red curve is the upper bound on the competitive ratio of any algorithm.

**Theorem 2.1.** *(Warmup) The greedy algorithm is* $0.5$*-competitive for online SWM.*

*Proof.* For each online item $i$, let $\alpha_i$ be the marginal gain by $Gr$ from allocating item $i$ upon its arrival. It is easy to see that $\sum_{i=1}^{n} \alpha_i$ is equal to the value of $Gr$. For each agent $j$, let $\beta_j$ be the total value of the allocation to $j$ at the end of the algorithm. By definition, we know that $\sum_{j=1}^{m} \beta_j$ is equal to the value of $Gr$. Thus, the value of $Gr$ is clearly $0.5 \left( \sum_{i=1}^{n} \alpha_i + \sum_{j=1}^{m} \beta_j \right)$.

Recall that $f_j(.)$ denotes the valuation function of agent $j$. Below, we show that $f_j(Opt_j)$ is upper-bounded by $\beta_j + \sum_{i \in Opt_j} \alpha_i$. Note that for distinct agents $j$ and $k$, $Opt_j$ and $Opt_k$ are disjoint. Thus, by summing over all agents, we can upper-bound the value of $Opt$ by $\sum_{i=1}^{n} \alpha_i + \sum_{j=1}^{m} \beta_j$. This means that the competitive ratio of $Gr$ is $0.5$.

Now, we just need to show that for any agent $j$ we have $f_j(Opt_j) \leq \beta_j + \sum_{i \in Opt_j} \alpha_i$. Note that for any item $i \in Opt_j$, the value of $\alpha_i$ is at least the marginal gain that would have been obtained from assigning $i$ to $j$ when it arrives. Applying submodularity of $f_j$, we have $\alpha_i \geq f_j(Gr_j \cup i) - f_j(Gr_j)$. Moreover, by definition we have $\beta_j = f_j(Gr_j)$. Thus, we have:

$$\beta_j + \sum_{i \in Opt_j} \alpha_i \geq f_j(Gr_j) + \sum_{i \in Opt_j} (f_j(Gr_j \cup i) - f_j(Gr_j))$$
$$\geq f_j(Gr_j) + \left( f_j(Gr_j \cup Opt_j) - f_j(Gr_j) \right)$$
$$= f_j(Gr_j \cup Opt_j) \geq f_j(Opt_j),$$

where the second inequality follows by submodularity, and the last inequality by monotonicity. This completes the proof. $\square$

**Lemma 2.2.** *Let $Gr_p$ be an algorithm that with probability $p$ passes each online item to $Gr$ for allocation, and leaves it unmatched otherwise. $Gr_p$ is $\frac{p}{1+p}$-competitive for online SWM.*

*Proof.* The proof here is fairly similar to Theorem 2.1. For each online item $i$, set $\alpha_i$ to be the marginal gain that would have been achieved from allocating item $i$ upon its arrival (assuming $i$ is passed to $Gr$), given the current allocation of items. Note that $\alpha_i$ is a random variable (depending on the outcome of previous decisions to pass items to $Gr$ or not), but it is independent of the coin toss that determines whether it is passed to $Gr$, and so the *expected* marginal gain of allocating item $i$, (given all previous allocations) is $pE[\alpha_i]$. Thus, by linearity of expectation, the expected value of $Gr_p$ is $pE[\sum_{i=1}^{n} \alpha_i]$. On the other hand, for each agent $j$, set $\beta_j$ to be the value of the *actual* allocations to $j$ at the end of the algorithm. Again, we have $\sum_{j=1}^{m} \beta_j$ equal to the value of $Gr_p$. Combining these two, we conclude that the expected value of $Gr_p$ is equal to $\frac{1}{1+1/p} \left( \sum_{i=1}^{n} E[\alpha_i] + \sum_{j=1}^{m} E[\beta_j] \right)$.

As before, we show that $f_j(Opt_j)$ is upper-bounded by $\beta_j + \sum_{i \in Opt_j} \alpha_i$. Therefore, we can conclude that the competitive ratio of $Gr_p$ is $\frac{1}{1+1/p} = \frac{p}{1+p}$.

It remains only to show that for any agent $j$, we have $f_j(Opt_j) \leq \beta_j + \sum_{i \in Opt_j} \alpha_i$. This is exactly the same as our proof for Theorem 2.1: By submodularity of $f_j$ we have, $\alpha_i \geq f_j(Gr_p(j) \cup i) - f_j(Gr_p(j))$, and by definition we have $\beta_j = f_j(Gr_p(j))$. We provide the complete proof in the full version. $\qquad\square$

The main theorem of this section follows immediately.

**Theorem 2.3.** *For any* $0 < p < 1$*, there is a* $(\frac{p}{1+p}, \frac{1-p}{2-p})$*-competitive algorithm for bi-objective online SWM.*

### 2.3  Hardness of Bi-Objective online SWM

We now prove that Theorem 2.3 is almost tight, by describing a hard instance for bi-objective online SWM. To describe this instance, we define notions of *super nodes* and *super edges*, which capture the hardness of maximizing a submodular function even in the offline setting. Using the properties of super nodes and edges, we construct and analyze a hard example for bi-objective online SWM.

Our construction generalizes that of Kapralov *et al.* [12], who prove the upper bound corresponding to the two points $(0.5, 0)$ and $(0, 0.5)$ in the curve shown in Figure 1. They use the following result: For any fixed $c_0$ and $\epsilon'$ it is NP-hard to distinguish between the following two cases for offline SWM with $n$ agents and $m = kn$ items. This holds even for submodular functions with 'coverage' valuations.

- There is an allocation with value $n$.
- For any $l \leq c_0$, no allocation allocates $kl$ items and gets a value more than $1 - e^{-l} + \epsilon'$.

Intuitively, in the former case, we can assign $k$ items to each agent and obtain value 1 per agent. In the latter case, even if we assign $2k$ items (however they are split across agents), we can obtain total value at most $0.865$. It also follows that there exist 'hard' instances such that there is an optimal solution of value $n$, but for any $l < 1$, any assigment of $ml$ edges obtains value at most $(1 - e^{-l} + \epsilon')n$.

We now define a *super edge* to be a hard instance of offline SWM as defined above. We refer to the set of agents in a super edge as the agent super node, and the set of items in the super edge as the item super node. If two super edges share a super node, it means that they share the agents / items corresponding to that super node in the same order. If (in expectation) we allocate $ml$ items of a super edge, we say the load of that super edge is $l$. Similarly, if (in expectation) we allocate $ml$ items to an agent super node, we say the load of that super node is $l$. Using the definition of super edge and super node, the hardness result of Kapralov *et al.* [12] gives us the following lemma:

**Lemma 2.4.** *Assume* $RP \neq NP$ *and let* $\epsilon$ *be an arbitrary small constant. If the (expected) load of a randomized polynomial algorithm on an agent super node is* $l$*, the expected welfare of all agents is at most* $(1 - e^{-l} + \epsilon)n$*.*

Now with Lemma 2.4 in hand, we are ready to present an upper bound for bi-objective online SWM.

**Theorem 2.5.** *Assume* $RP \neq NP$*. The competitive ratio* $(\alpha, \beta)$ *of any algorithm for bi-objective online SWM is upper bounded by the red curve in Figure 1. More precisely (assuming w.l.o.g. that* $\alpha \geq \beta$*), for any* $\gamma \in [0, 1]$*, there is no algorithm with* $\alpha > \frac{0.5 + \gamma^2/6}{1 + \gamma^2}$ *and* $\beta > \gamma\alpha$*.*

## 3  Bi-Objective Online Weighted Matching

In this section, we consider two special cases of bi-objective online SWM, each of which generalizes the (single objective) online weighted matching problem (with free disposal). Here, each item $i$ has two weights $w_{ij}^f$ and $w_{ij}^g$ for agent $j$, and each agent $j$ has (large) capacity $C_j$. The weights of item $i$ are revealed when it arrives, and the algorithm must allocate it to some agent immediately.

In the *first model*, after the algorithm terminates, and each agent $j$ has received items $\mathcal{S}_j$, it chooses a subset $\mathcal{S}_j' \subseteq \mathcal{S}_j$ of at most $C_j$ items. The total value in the first objective is then $\sum_j \sum_{i \in \mathcal{S}_j'} w_{ij}^f$, and in the second objective $\sum_j \sum_{i \in \mathcal{S}_j'} w_{ij}^g$. Intuitively, each agent must pick a subset of its items, and it

**Exponential Weight Algorithm.**
Set $\beta_j$ to 0 for each agent $j$.
Upon arrival of each item $i$:

    1. If there is agent $j$ with $w_{ij} - \beta_j > 0$

        (a) Let $j$ be the agent that maximizes $w_{ij} - \beta_j$

        (b) Assign $i$ to $j$, and set $\alpha_i$ to $w_{ij} - \beta_j$.

        (c) Let $w_1, w_2, \ldots, w_{C_j}$ be the weights of the $C_j$ highest weight items, matched to $j$ in a non-increasing order.

Figure 2: Exponential weight algorithm for online matching with free disposal.

gets paid its (additive) value for these items. In the (single-objective) case where each agent can only be *allocated* $C_j$ items, this is the online weighted $b$-matching problem, where vertices are arriving online, and we have edge weights in the bipartite (item, agent) graph. This problem is completely intractable in the online setting, while the free disposal variant [9] in which additional items can be assigned, but at most $C_j$ items count towards the objective, is of theoretical and practical interest.

In the *second model*, after the algorithm terminates and agent $j$ has received items $\mathcal{S}_j$, it chooses two (not necessarily disjoint) subsets $S_j'^f$ and $\mathcal{S}_j'^g$; items in $\mathcal{S}_j'^f$ are counted towards the first objective, and those in $\mathcal{S}_j'^g$ are counted towards the second objective.

**Theorem 3.1.** *For any $(\alpha, \beta)$ such that $\alpha + \beta \leq 1 - \frac{1}{e}$, there is an $(\alpha, \beta)$-competitive algorithm for the first model of the bi-objective online weighted matching. For any constant $\epsilon > 0$, there is no such algorithm when $\alpha + \beta > 1 - \frac{1}{e} + \epsilon$.*

To obtain the positive result, with probability $p$, run the exponential weight algorithm (see Figure 2) for the first objective (for all items), and with probability $1 - p$ run the exponential weight algorithm for the second objective for all items; this combination is $(p(1 - \frac{1}{e}), (1 - p)(1 - \frac{1}{e}))$-competitive. We deffer the proof of this and the matching hardness results to the full version.

Having given matching upper and lower bounds for the first model, we now consider the second model, where if we assign a set $\mathcal{S}_j$ of items / edges to an agent $j$ we can select two subsets $\mathcal{S}_j'^f, \mathcal{S}_j'^g \subseteq \mathcal{S}_j$ and use them for the first and second objective functions respectively.

**Theorem 3.2.** *There is a $(p(1 - \frac{1}{e^{1/p}}), (1 - p)(1 - \frac{1}{e^{1/(1-p)}}))$-competitive algorithm for the bi-objective online weighted matching problem in the second model as $\min_j\{C_j\}$ tends to infinity.*

**Theorem 3.3.** *The competitive ratio of any algorithm for bi-objective online weighted matching in the second model is upper bounded by the curve in figure 3.*

## 4 Bi-Objective Online Budgeted Allocation

In this section, we consider the bi-objective online allocation problem where one of the objectives is a budgeted allocation problem and the other objective function is weighted matching. Here, each item $i$ has a weight $w_{ij}$ and a bid $b_{ij}$ for agent $j$. Each agent $j$ has a capacity $C_j$ and a budget $B_j$. If an agent is allocated items $\mathcal{S}_j$, for the first objective (weighted matching), it chooses a subset $\mathcal{S}_j'$ of at most $C_j$ items; its score is $\sum_{i \in \mathcal{S}_j'} w_{ij}$. For the second objective, its score is $\min\{\sum_{i \in \mathcal{S}_j} b_{ij}, B_j\}$. Note that in the second objective, the agent does not need to choose a subset; it obtains the sum of the bids of all items assigned to it, capped at its budget $B_j$.

Clearly, if we set all bids $b_{ij}$ to 1, the goal of the budgeted allocation part will be maximizing the cardinality. Thus, this is a clear generalization of the bi-objective online allocation to maximize weight and cardinality, and the same hardness results hold here.

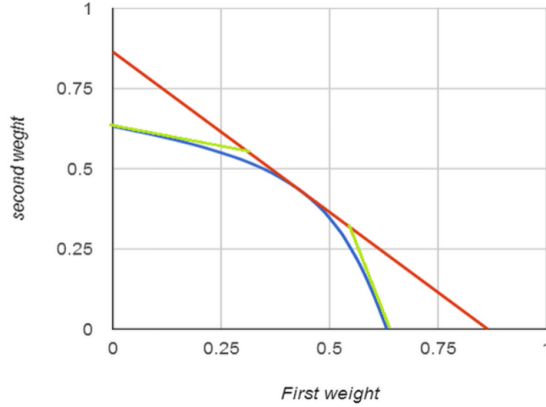

Figure 3: The blue curve is the competitive ratio of our algorithm in the second model, while the red line and the green curves are the upper bounds on the competitive ratio of any algorithm.

As is standard, throughout this section we assume that the bid of each agent for each item is vanishingly small compared to the budget of each bidder. Interestingly, again here, we provide a $(p(1 - \frac{1}{e^{1/p}}), (1-p)(1 - \frac{1}{e^{1/(1-p)}}))$-competitive algorithm, which is almost tight. At the end, as a corollary of our results, we provide a a $(p(1 - \frac{1}{e^{1/p}}), (1-p)(1 - \frac{1}{e^{1/(1-p)}}))$-competitive algorithm, for the case that both objectives are budgeted allocation problems (with separate budgets).

Our algorithm here is roughly the same as for two weight objectives. For each item, with probability $1-p$, we pass it to the Exponential Weight algorithm for matching, and allocate it based on its weight. With the remaining probability $p$, we assign the algorithm based on its bids and count it towards the Budgeted Allocation objective. However, the algorithm we use for Budgeted Allocation is slightly different: We *virtually* run the Balance algorithm of Mehta *et al.* [17] for Budgeted Allocation (Fig. 4), as though we were assigning *all* items (not just those passed to this algorithm), but with each item's bids scaled down by a factor of $p$. For those $p$ fraction of items to be assigned by the Budgeted Allocation algorithm, assign them according to the recommendation of the virtual Balance algorithm.

Theorem 3.2 from the previous section shows that our algorithm is $(1-p)(1 - \frac{1}{e^{1/(1-p)}})$-competitive against the optimum weighted matching objective. Thus, in the rest of this section, we only need to show that this algorithm is $p(1 - \frac{1}{e^{1/p}})$-competitive against the optimum Budgeted Allocation solution. First, using a primal dual approach, we show that the outcome of the virtual Balance algorithm (that runs on $p$ fraction of the value of each item) is $p(1 - \frac{1}{e^{1/p}})$ against the optimum with the actual weights. Then, using the Hoeffding inequality, we show that the expected value of our allocation for the budgeted allocation objective is fairly close to the virtual algorithm's value, i.e. the difference between the competitive ratio of our allocation and the virtual allocation is $o(1)$.

**Lemma 4.1.** *When $\frac{\max_{i,j} b_{ij}}{B_j} \to 0$, the total allocation of the virtual balance algorithm that runs on $p$ fraction of the value of each bid is at least $p(1 - \frac{1}{e^{1/p}})$ times that of the optimum with the actual values.*

The proof of this lemma is similar to the analysis of Buchbinder *et al.* [3] for the basic Budgeted Allocation problem. We provide this proof in the full version.

**Lemma 4.2.** *For any constant $p$, assuming $\frac{\max_{i,j} b_{ij}}{B_j} \to 0$, the budgeted allocation value of our algorithm tends to the value of Balance with $p$ fraction of each bid, with high probability.*

In the virtual Balance algorithm, we allocate $p$ fraction of *each* item, while in our real algorithm, we allocate every item according to the virtual Balance algorithm with probability $p$. Since each item's bids are small compared to the budgets, the lemma follows from a straightforward concentration argument. We present the complete proof in the full version.

The following lemma is an immediate result of combining Lemma 4.1 and Lemma 4.2.

---

**Virtual Balance algorithm on $p$ fraction of values.**
Set $\beta_j$ and $y_j$ to 0 for each agent $j$.
Upon arrival of each item $i$:

    1. If $i$ has a neighbor with $b_{ij}(1 - \beta_j) > 0$

        (a) Let $j$ be the agent that maximizes $b_{ij}(1 - \beta_j)$

        (b) Assign $i$ to $j$ i.e. set $x_{ij}$ to 1.

        (c) Set $\alpha_i$ to $b_{ij}(1 - \beta_j)$.

        (d) Increase $y_j$ by $\frac{b_{ij}}{B_j}$

        (e) Increase $\beta_j$ by $\frac{e^{y_j - 1/p}}{1 - e^{-1/p}} \frac{b_{ij}}{B_j}$

    2. Else: Leave $i$ unassigned.

---

Figure 4: Maintaining solution to primal and dual LPs.

**Lemma 4.3.** *For any constant p, assuming $\frac{max_{i,j} b_{ij}}{B_j} \to 0$, our algorithm is $p(1 - \frac{1}{e^{1/p}})$-competitive against the optimum budgeted allocation solution.*

Lemma 4.3 immediately gives us the following theorem.

**Theorem 4.4.** *For any constant p, assuming $\frac{max_{i,j} b_{ij}}{B_j} \to 0$, there is a $(p(1 - \frac{1}{e^{1/p}}), (1 - p)(1 - \frac{1}{e^{1/(1-p)}}))$-competitive algorithm for the bi-objective online allocation with two budgeted allocation objectives.*

Moreover, if we pass each item to the exponential weight algorithm with probability $p$, the expected size of the output matching is at least $p(1 - \frac{1}{e^{1/p}})$ that of the optimum [14]. Together with Lemma 4.3, this gives us the following theorem.

**Theorem 4.5.** *For any constant p, assuming $\frac{max_{i,j} b_{ij}}{B_j} \to 0$, there is a $(p(1 - \frac{1}{e^{1/p}}), (1 - p)(1 - \frac{1}{e^{1/(1-p)}}))$-competitive algorithm for the bi-objective online allocation with a budgeted allocation objective and a weighted matching objective.*

## 5 Conclusions

In this paper, we gave the first algorithms for several bi-objective online allocation problems. Though these are nearly tight, it would be interesting to consider other models for bi-objective online allocation, special cases where one may be able to go beyond our hardness results, and other objectives such as fairness to agents.

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
