[Reviews · NeurIPS 2016]

Reviewer 1

Summary

The paper studies the online submodular welfare maximization problem with two objectives. In this setting, there are two monotone submodular objective functions f and g on a ground set I. The items of I arrive online and, when an item arrives, it needs to be allocated to a single agent j. The goal is to maximize the submodular welfare in both objectives, that is, maximize sum_j f(A_j) and sum_j g(A_j), where A_j is the set of items allocated to agent j. The main contributions of the paper are tight approximation guarantees for this problem and some of its special cases; the approximation/competitive ratio of an algorithm is (alpha, beta) if the allocation is an alpha-approximation to the submodular welfare problem with respect to f and a beta-approximation to the submodular welfare problem with respect to g. The main algorithm for general objective functions is a very natural algorithm that runs the Greedy algorithm for two instances of the single objective submodular welfare maximization problem, and when an item arrives it allocates to one of the instances randomly with probability p. Using a very nice argument, the authors show that this algorithm achieves a (p/(1 + p), (1 - p)/(2 -p)). Interestingly, this competitive ratio is best possible, and this matching lower bound builds on techniques from previous work.

Qualitative Assessment

The paper makes the first contributions to the online submodular welfare maximization problem with two objectives. The theoretical results are novel, and they provide matching upper and lower bounds on the competitive ratio, thus settling the approximability of the problem. The algorithms are very natural, and the analyses are insightful. == post-rebuttal answer== I have read the entire rebuttal.

Confidence in this Review

2-Confident (read it all; understood it all reasonably well)


Reviewer 2

Summary

The paper builds on the following problem. There is a set of agents. Each agent a is associated a submodular function f_a. Elements from the universe arrive over time. When an element arrives, it is to be assigned to an agent. Let S_a be the set of elements assigned to agent a. The goal is to assign the elements to agents as to maximum sum_a f(S_a). This problem is known as online submodular welfare maximization. For this problem, the greedy algorithm which greedily assigns elements to the agent that maximizes the objective is a 1/2 approximation and this is the best possible. This paper considers a generalization of this problem where each agent has two different submodular functions. These two functions represent two different objectives each agent may want to optimize. The goal is to design an an allocation of elements so as to give a (a,b)-approximation where 'a' is the approximation ratio if only the first function is considered in the objective and similarly 'b' is for the second function. The paper shows what is essentially a (1/3,1/3) approximation for the problem. This is done simply by flipping a coin when an element arises and choosing to runt the greedy algorithm only considering one of the objectives. Proving this result is straightforward. What is nice about this work is they show that this is the best possible algorithm but giving a lower bound. This I found interesting. The paper goes on to show better results in some restricted cases. I think this is a reasonable paper. The downside of the work is that they did not motivate the two objective setting much and this leaves my feeling that it is a contrived. On the other hard, I personally think the problem is reasonable despite the lack of motivation given. I feel the authors add insight into this area. I feel the paper is a weak accept.

Qualitative Assessment

Please give more detailed motivation for the problem if available.

Confidence in this Review

2-Confident (read it all; understood it all reasonably well)


Reviewer 3

Summary

The authors analyze online submodular optimization where the designer has two objectives. They give upper and lower bounds on approximation ratios achievable in this bi-criterion approximation problem. They show that simple greedy algorithms with a random selection criterion achieve tight approximation ratios.

Qualitative Assessment

I found the paper interesting and the question relevant. The biggest plus of the paper is the tightness of their result and the fact that the tight results are achievable by simple algorithms. However, the latter is also a caveat as the algorithms and techniques are not that technically interesting and only lead guarantees in expectation rather than high probability ones.

Confidence in this Review

2-Confident (read it all; understood it all reasonably well)


Reviewer 4

Summary

The paper considers the problem of optimal online allocation when optimality is defined by two submodular functions. The authors first provide general results on the existence of algorithms with a certain competitiveness guarantee over the two objective functions. They then consider two well known problems from online advertisement (online display ad allocations, and the AdWords problem), and extend them to a bi-objective case. There they show how to extend known greedy algorithms to the bi-objective case, and show them to be near optimal, with (almost) matching lower and upper bounds. The setting considered by the authors is of great practical relevance for consumer-facing internet industries (such as e-commerce and online advertising). There, it is common to try to optimize non-correlated metrics, such as “share of voice” and “revenue over ad-spend”.

Qualitative Assessment

My assessment of the potential impact is driven by the mismatch of the paper to the community. In the right community, this paper is likely going to be very impactful.

Confidence in this Review

3-Expert (read the paper in detail, know the area, quite certain of my opinion)


Reviewer 5

Summary

The paper introduces multi-objective version of submodular optimization problems and proposes randomized algorithms that have guaranteed competitive ratio. It also provides (nearly/exact) matching hardness results.

Qualitative Assessment

The paper is well written. The idea (select f/g with prob p/(1-p)) is very simple, but seems to be useful in many other similar problems (e.g., three or more objective functions). It would be more better to include experimental results.

Confidence in this Review

3-Expert (read the paper in detail, know the area, quite certain of my opinion)


Reviewer 6

Summary

The paper focuses on the online allocation problems, which is formulated via bi-objective online sub-modular optimization. Upper and lower bounds on allocation items to targeted agents are derived.

Qualitative Assessment

The paper is technically sound. I could not spot technical flaws (have not checked the proofs though). However, there is a great deal of research done on matching and allocation, of which I am not fully aware, hence I cannot fully judge the added value the paper in terms of novelty. Even though the paper is motivated merely by an important application, the paper lacks the relevant simulations.

Confidence in this Review

1-Less confident (might not have understood significant parts)